# Evaluation of a diagnostic device, CL *Detect* rapid test for the diagnosis of new world cutaneous leishmaniasis in Peru

**Max Grogl**[1]☯**, Christie A. Joya**[2]☯*, **Maria Saenz**[3‡], **Ana Quispe**[4‡], **Luis Angel Rosales**[3‡], **Rocio del Pilar Santos**[3‡], **Maxy B. De los Santos**[2‡], **Ngami Donovan**[5‡], **Janet H. Ransom**[5‡], **Ana Ramos**[4‡], **Elmer Llanos Cuentas**[4]☯

**1** Naval Medical Research Center, Silver Spring, Maryland, United States of America, **2** Department of Parasitology, U.S Naval Medical Research Unit No. 6, Callao, Lima, Peru, **3** Vysnova Partners, North Englewood, Maryland, United States of America, **4** Universidad Cayetano Heredia, Lima, Peru, **5** Fast-Track Drugs & Biologics, LLC, Poolesville, Maryland, United States of America

☯ These authors contributed equally to this work.
‡ MS, AQ, LAR, RDPS, MBDLS, ND, JHR and AR also contributed equally to this work.
* christie.a.joya.mil@health.mil

**Data Availability Statement:** All relevant data are within the manuscript and its Supporting Information files.

## Abstract

### Background

Cutaneous leishmaniasis (CL) is a neglected disease and a public health problem in Latin America. The diagnosis of CL in poor hyperendemic regions relies to large extent on the identification of amastigotes in Giemsa-stained smears. There is an urgent need for a rapid, sensitive and low cost diagnostic method for use in field conditions for CL as current modalities are not readily available. The primary objective of this study was to determine the sensitivity and specificity of the FDA-cleared CL *Detect* Rapid Test in Peru, using modified test procedures rather than the instructions-for-use, by 1) increasing the extraction time and 2) increasing the volume of the sample added to the test strip. CL *Detect* Rapid Test results were compared against microscopy and kDNA-PCR, for the diagnosis of CL in ulcerated lesions. In addition, we compared two collection methods the dental broach used and mentioned in the CL Detect insert and the standard less invasive and easier to conduct scrapping method.

### Methodology

Participants were patients who presented for medical consultation due to a suspected CL lesion. Four samples from the index lesion were collected using a dental broach, per package insert, and lancet scraping and tested by the modified CL *Detect* Rapid Test, microscopy, and PCR.

### Principal findings

A total of 156 subjects were eligible and evaluated. The modified CL *Detect* sensitivity was higher in specimens obtained by scraping (83.3%) than those from dental broach (64.2%). The specificity was lower in scrapings (77.8%) with a false positive rate of 22.2% compared

**Funding:** MG received support by funding from the US Department of Defense Health Agency 6.7 Intramural Funding, https://www.health.mil/About-MHS/OASDHA/Defense-Health-Agency. The funders had no role in study design, data collection and analysis, decision to publish, or preparation of the manuscript.

**Competing interests:** The authors have declared that no competing interests exist.

with dental broach samples (91.7%) with a false positive rate of 8.3%. However, molecular analysis showed that all 8 false negative microscopy scrapings (those positive by modified CL Detect and negative by microscopy) were positive by kDNA-PCR, meaning that the modified CL Detect was more sensitive than microscopy.

## Conclusions

These modifications to the package insert that resulted in a diagnostic sensitivity (83.3%) comparable to microscopy for species found in Peru may enable earlier anti-leishmanial drug treatment decisions based on a positive result from the CL *Detect* Rapid Test alone until further diagnostic tests like microscopy and PCR can be performed.

## Trial registration

NCT03762070; Clinicaltrials.gov.

## Author summary

Cutaneous leishmaniasis is commonly referred to as a neglected tropical disease. Presented with many species, the ones existing in Peru are rarely deadly, but it can cause chronic non-healing ulcers and scarring that can lead to contractures and social stigma. Due to the toxicity of the treatments for this disease, most ministries of health require a diagnosis prior to its initiation. Unfortunately, diagnosis can be quite difficult, as the gold standard of microscopy requires highly trained personnel. As the use of microscopy, the rapid, point-of-care diagnostic device CL *Detect* has been a viable option for Old World cutaneous leishmaniasis; however, in the New World the FDA-cleared device has low sensitivity. In this study, three simple changes (sample collection by the standard method of scraping using a lancet, increased extraction time, and increased sample volume) to the instructions were implemented and showed a similar diagnostic sensitivity to microscopy for the diagnosis of cutaneous leishmaniasis in Peru.

## Introduction

Cutaneous leishmaniasis (CL) is a parasitic disease that causes disfiguring lesions. It is mainly found in the tropical and subtropical areas in the Middle East, southwest Asia, the Mediterranean coast, sub-Saharan Africa, Mexico, and Central and South America, and it is spread through the bite of infected sand flies. It places over 1 billion people at risk worldwide, with 1.5 million new cases emerging annually, including U.S. service members serving abroad [1,2].

The CL *Detect* Rapid Test is a qualitative, *in vitro* immunochromatographic assay for the rapid detection of *Leishmania* species antigen in ulcerative skin lesions. The test is intended for use with dental broach samples from ulcerative skin lesions that are obtained from patients with suspected cutaneous leishmaniasis. The test targets the peroxidoxin antigen of *Leishmania* species that may cause CL and is intended to aid in the diagnosis of CL as it must be interpreted within the context of relevant epidemiologic, clinical and laboratory findings [3].

The *CL Detect* was developed by the U.S. Army in partnership with InBios International Inc. Seattle, WA, United States of America, the Institut Pasteur de Tunis in Tunisia, Walter Reed Army Institute of Research, and the U.S. Army Small Business Innovation Research

Program (SBIR). This program allowed InBios, a small, U.S. medical device manufacturer, the opportunity to provide an innovative research and development solution in response to a critical Army need; the rapid diagnosis of cutaneous leishmaniasis in an operational setting. It received clearance from the Food and Drug administration (FDA) in 2014.

*CL Detect* works when freely circulating antigens in the sample react with a colloidal gold-conjugate of a monoclonal antibody to Thiol-specific antioxidant (TSA) [4]. These two entities will form an antigen-antibody complex that will flow vertically up the test stick until encountering an immobilized detection zone (the TSA test line) containing an unconjugated rabbit polyclonal antibody to TSA. At the TSA test line, an accumulation of color from the colloidal gold will indicate a positive sample, denoting the presence of *Leishmania* parasites in the lesion sample.

The approval of the *CL Detect* Rapid Test was based on clinical trials in Tunisia (Old World CL species) and for specificity in the United States (ulcerated lesions caused by other etiologies) [3,5]. However, the sensitivity of diagnostic tests is known to depend on the species of *Leishmania* and on the amount of amastigotes (a thiol-specific antioxidant source, TSA antigen) found in the CL lesion [4,6]. In Kabul Afghanistan a major focus of CL caused by *Leishmania tropica*, the CL Detect Rapid Test had a 65.4% sensitivity [95% Confidence Interval (CI): 59.2–71.2%] and 100% specificity [95% CI: 80.5–100%] [7]. In rural areas in Morocco CL is mainly caused by *Leishmania* major (southeast of the country) and *Leishmania tropica* central and northern regions. The CL Detect Rapid Test sensitivity in Morocco was 68% [95% confidence interval (CI): 61–74], specificity 94% [95% CI: 91–97], positive predictive value 95% [95% CI: 92–98], and negative predictive value 64% [95% CI: 58–70] [8]. Meanwhile in Ethiopian patients suspected of CL caused by *Leishmania aethiopica* the sensitivity of the CL Detect Rapid Test on the skin slit was 31.3% (95% confidence interval (CI) 23.9–39.7), which was significantly higher (p = 0.010) than for the dental broach (22.7%, 95% CI 16.3–30.6) [9]. Differences in sensitivity across endemic regions might be explained by either differences in the levels of the target antigen peroxidoxin at the species levels or in its sequence or confirmation The parasite load of lesions from Old World species (*Leishmania major*) tend to be higher than those found in the New World (example *L. peruviana* and *L. braziliensis* infections) [10–14]. Furthermore, Fig 1 shows that the color intensity on the TSA test line of the CL *Detect* Rapid Test strip correlates with the number of promastigotes, as the TSA antigen is present in both amastigotes and promastigotes [3]. Thus, it is reasonable to believe that the sensitivity of the FDA-cleared CL *Detect* Rapid Test can be lower in New World *Leishmania* endemic regions (like Peru) where the number of parasites in the tissue is lower compared to Tunisia [11].

To expand the diagnostic options for CL, the CL *Detect* Rapid Test (CL Detect) and the Loopamp *Leishmania* Detection Kit (Loopamp) were evaluated in Surinam where CL is a serious health problem [12]. Not surprisingly, the study showed that the CL Detect had a very low sensitivity compared to microscopy (36.7%) or PCR (35.8%), due to a high number of false negative results. Thus, the authors concluded CL Detect was not likely to be a good replacement for the routine diagnostic procedure for CL in Suriname [15].

Likewise, in a 2014–2016 study conducted by NAMRU-6 and the Instituto de Medicina Tropical Alexander von Humboldt (IMTAvH), in Peru (PERU-01) we found the sensitivity of the CL *Detect* Rapid Test was only 50% using the device instructions.

In the present study (PERU-02), and based on the published scientific evidence about the CL Detect Rapid Test (Afghanistan, Morocco, Ethiopia) we hypothesize that implementing two simple modifications of the test procedure described in the device instructions the sensitivity of the FDA-cleared CL *Detect* Rapid Test could increase and it will suffice to deliver a diagnosis in a short time, accelerating patient's access to treatment in remote endemic areas

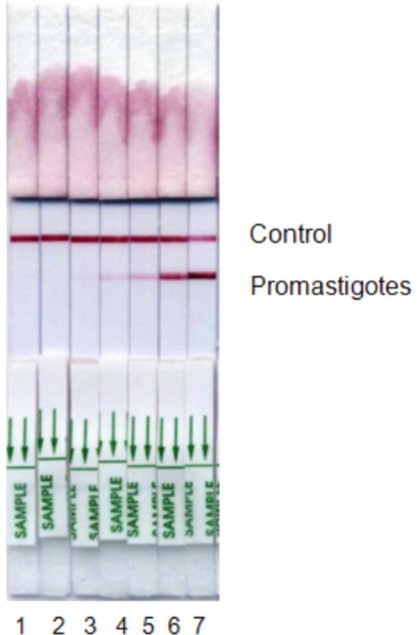

**Fig 1. CL *Detect* Test Strip Intensity Compared to Number of Promastigotes.** The intensity of the TSA Test Line increases with increasing parasite number. Lanes 1–7 show *Leishmania major* promastigotes from 0, 10, 100, 500, $10^3$, $10^4$, $10^5$ parasites per assay (20 µl sample).

and real field situations. Additionally, we compared sample taking with the dental broach (per package insert) with the traditional lancet scraping to evaluate its viability, as microscopy is the most common collection method used in the field.

## Methods

### Ethics statement

The study protocol (NAMRU6.2015.0009) was reviewed and approved by the Institutional Review Boards (IRBs) of the U.S Naval Medical Research Unit No. 6 in compliance with all applicable federal regulations governing the protection of human subjects, and Universidad Peruana Cayetano Heredia and received approval from the Regional Health Directorate of Madre de Dios and from the Ministry of Health of Peru Regulatory Office (OGITT-INS) and registered at Clinicaltrials.gov (NCT03762070). Written informed consent was obtained from all study participants. This trial was conducted in accordance with the ethical principles of Good Clinical Practices (GCP), according to the International Conference on Harmonization (ICH) Harmonized Tripartite Guideline.

### Study population

Adults ($\geq$18 years of age) presenting to Universidad Peruana Cayetano Heredia (UPCH) and the regional health posts and centers in the Madre de Dios region of Peru (Jorge Chávez, Milenio, Iberia and Santa Rosa) for evaluation of skin lesions consistent with CL were invited to participate in the study. Inclusion criteria included an index lesion that was primarily

ulcerative in appearance (not purely verrucous or nodular), present for less than 2 months, without clear evidence of cellulitis and located in a suitable area for collecting samples by both dental broach and lancet scraping, as well as being capable of understanding and complying with the protocol. Subjects were excluded if they had received anti-leishmanial treatment in the 2 months prior to presentation or if the lesion had evidence of manipulation. Enrollment took place between October 2018 and September 2019, to meet the predefined sample size.

## Sample collection

After obtaining informed consent and screening participants for eligibility, an index lesion was identified and sampled on the internal lesion border in the following order: 1) dental broach sample for modified CL *Detect* Rapid testing; 2) stainless steel, sterile lancet sample for microscopy to assess the presence of *Leishmania* amastigotes; 3) lancet sample for PCR; and 4) when possible (depending on the size of the lesion), a sample from a different site of the same lesion, obtained by scraping by lancet for use with the modified CL *Detect* Rapid Test. Adverse events (AEs) and unanticipated adverse device events (UADEs) were collected in the clinic starting with the time of the collection of the first lesion sample with the dental broach and until the subject left the clinic after study procedures were completed.

Lancet scraping was performed as follows, the lesion of the patient was cleaned using an alcohol gel pad, then the crust was removed with the blunt end of a stainless-steel sterile lancet and the tissue was obtained from the internal border of the lesion.

Participant demographics including age, gender, and ethnicity were collected. The length of time between initial presence of index lesion and time of lesion sampling, history of leishmaniasis including any prior (past 2 months) or current treatments for leishmaniasis, current medication use (name of medication, dose, and start date), and the anatomical location of the lesion were also captured.

## Modified CL *Detect* rapid test

The following modifications to the device instructions were used: 1) increase extraction time in lysis buffer from 5 to 10 minutes to 20 to minutes; and 2) increase from 20 μL to 40 μL in the volume of the sample to add to the test strip in the area beneath the arrow. Lesion tissue obtained via dental broach sampling was placed in the lysis buffer for 20 minutes but not more than 30 minutes, then 40 μL directly applied onto the reservoir pad of the CL *Detect* Rapid Test dipstick. The scraping taken from a different site of the same lesion was also tested with the modified CL *Detect* Rapid Test.

Satisfactory performance of the assay was confirmed by the appearance of color at the control line. The result of the test was recorded and photographed. The intensity of the "signal" (color at the TSA test line) was compared with a color intensity chart (scale 0 to 15) provided by InBios and the results were recorded.

## Microscopy

Microscopy slides were prepared at the enrolling sites, smears were made by spreading the scraping material to make a thin preparation on two separate slides that were air dried, fixed with methanol and stained with Giemsa solution or Diff-Quick stain. The presence of parasites in the index lesion was determined by performing parasite counts on scraping samples by clinical staff at UPCH. Amastigotes in macrophages were counted according to the World Health Organization (WHO) guidance for grading parasite density using a 10x eyepiece and 100x oil objective lens [16]. The results were not shared with the enrolling sites laboratory personnel conducting the CL *Detect* test. A third slide was prepared and retained at the NAMRU-6

Puerto Maldonado site for diagnosis of the patients who participated in the study there. Only amastigotes with clearly defined and correctly stained nuclei, kinetoplasts, and cytoplasm and cell membranes were counted. The average parasite density was then determined according to the WHO reference standard and the results were recorded.

### *Leishmania* diagnostic methodology by kDNA-PCR

DNA extraction was carried out using the DNeasy Blood & Tissue Kit (Qiagen) following the manufacturer recommendations. DNA detection by kDNA-PCR was performed using 4 μL DNA and following the method targeting a region of *Leishmania* minicircle for amplification of CL-causing species as described elsewhere [17]. This PCR was used as a highly sensitive molecular test to compare to the performance of CL *Detect* Rapid Test and microscopy. The PCR assay was performed by NAMRU-6 Lima laboratory technicians. These personnel were blinded, did not have access to the results generated by either the enrolling sites laboratory personnel nor the microscopists trained for the study.

### Determination of *Leishmania* species by FRET-based Nested Real-Time PCR

For species identification, a FRET-based Nested Real-Time PCR targeting the mannose phosphate isomerase (MPI) and 6-phosphogluconate dehydrogenase (6PGD) genes was performed and melting peaks patterns were compared with those from reference strains *L. (V.) braziliensis* (MHOM/BR/84/LTB300), *L. (V.) peruviana* (MHOM/PE/87/PAB2880), *L. (V.) guyanensis* (MHOM/BR/75/M4147), *L. (V.) panamensis* (MHOM/PA/71/LS94), *L. (V.) lainsoni* (MHOM/PE/88/BAB1730), *L. (V.) amazonensis* (MHOM/BR/73/M2269) and *L. (V.) mexicana* (MHOM/BZ/82/BEL21) following conditions previously established [18]. Briefly, a 50 μL reaction was prepared containing of 1X Taq polymerase buffer (Invitrogen), 1.5 mM MgCl$_2$, 200 μM dNTPs (Invitrogen), 0.8 μM or 1 μM of each primer (6PGD and MPI, respectively), 1.5 units of Taq DNA polymerase (Invitrogen), and 5 μL of DNA sample. Amplification conditions consisted of an initial denaturation at 94˚C for 5 min followed by 35 cycles of denaturation at 94˚C for 45 sec, annealing at 57˚C (for MPI) or 62˚C (for 6PGD) for 45 sec, and extension at 72˚C for 90 sec; and a final extension at 72˚C for 7 min for MPI or 5 min for 6PGD.

The second reaction was carried out in a 20 μL reaction for each gene containing 1X LightCycler 480 Genotyping Master (Roche, Indianapolis, IN), 1.25 μM of forward primer, 0.25 μM of reverse primer, 0.75 μM of anchor and sensor probes, and 5 μL of PCR product from the first reaction.

The amplification setting was performed on a LightCycler 480 and included an initial denaturation at 95˚C for 5 min followed by 45 cycles of denaturation at 95˚C for 10 sec, annealing at 60˚C for 20 sec under a single acquisition step) and extension at 72˚C for 20 sec. A melting curve analysis was performed at the end of the amplification cycles by heating the amplicons at 95˚C for 10 sec, cooling at 50˚C for 59 sec and then gradually increasing the temperature to 80˚C with one acquisition step each ˚C.

### Statistical methods

Summary statistics including number and percentage for discrete variables, and number, mean, standard deviation, median, minimum, and maximum values for continuous variables, and by-participant displays of all data collected were estimated. The evaluable analysis set included all subjects for whom the modified CL *Detect* Rapid Test and microscopy was performed and a result was reported as either positive or negative. The safety analysis set included all subjects who underwent any baseline procedures on Study Day 1 including preparation of

the lesion site and sample collection from the index lesion. The performance of modified CL *Detect* Rapid Test (sensitivity and specificity) was compared with the gold standard, microscopy of a stained lesion sample for identification of *Leishmania* amastigotes. Sensitivity was calculated as the number of true positives divided by the sum of the number of true positives plus the number of false negatives. Specificity was calculated as the number of true negatives divided by the sum of the number of true negatives plus the number of false positives (full methods in **S1 File**).

### Primary diagnostic analysis

The performance of modified CL *Detect* Rapid Test (sensitivity and specificity) along with 2-sided exact binomial 95% CI was determined by comparing with the gold standard, microscopy of stained lesion samples for identification of *Leishmania* amastigotes. This was done using dental broach and scraping by lancet samples separately for all analyses.

### Determination of sample size

The sample size determination of sensitivity at this site in Peru was based on the prior pilot study in Peru using the modified procedure, it was assumed that the point estimate for test sensitivity will be at least 80% (with a false negative rate = 20%). Using the approach for determining samples sizes for diagnostic tests by Malhotra, the minimum N that will have a lower 95% CI that is least 70% was estimated in 90 subjects [19].

We report the results using the Standards for the Report of Diagnostic accuracy studies (STARD) (S2 and S3 Files).

## Results

### Clinical features

Written consent was obtained from a total of 158 subjects. All but two (n = 156, 98.7%) of the consented subjects were eligible, underwent all testing procedures, and completed the study; zero participants terminated early.

Of the 156 subjects who participated in the study, 83.3% were male, the median age was 31.5 years (range 18 to 84), and all but one was Hispanic or Latino (Table 1). Most lesions were found on the legs (41.0%) and arms (23.1%) (Table 1). Per kDNA-PCR analysis, only 3 lesions were negative for leishmaniasis, and of the 153 positive, 67.9% were *L. (V.) braziliensis*, 13.5% were *L. (V.) peruviana*, 9.6% could not be identified by FRET-based Nested Real-Time PCR and the remaining were *L. (V.) guyanensis* (4.5%), and *L. (V.) lainsoni* (2.6%). The mean age of lesions from the time that they were first noticed by the study subject to the time of testing was 35.7 days (range 3 to 61 days).

### Sensitivity and specificity of modified CL *Detect* vs microscopy

All samples were collected from a single lesion using a dental broach and sterile lancet. Of the 156 specimens, 120 (77%) were positive and 36 (23%) were negative by microscopy. Of the microscopy positive samples, 77 were also positive by modified CL *Detect* Rapid Test when the sample was collected with a dental broach but 100 were positive by modified CL *Detect* Rapid Test when the sample was collected by scraping using the blunt end of the lancet (Table 2). Of the 36 samples negative by microscopy, 33 were also negative by modified CL *Detect* Rapid Test when the sample was collected with a dental broach but fewer, 28, were negative by modified CL *Detect* Rapid Test when the sample was collected by scraping-lancet (Table 2).

**Table 1. Demographic and Other Baseline Characteristics.**

| Demographics–Evaluable Subjects (N = 156) | |
|---|---|
| **Sex** | |
| Female, N (%) | 26 (16.7) |
| Male, N (%) | 130 (83.3) |
| **Age (years at date of consent)** | |
| Mean (SD) | 35.3 (14.5) |
| Median | 32 |
| Range | 18–84 |
| **Ethnicity-Evaluable subjects** | |
| Hispanic or Latino, N (%) | 155 (99.4) |
| Not Hispanic or Latino, N (%) | 1 (0.6) |
| **Baseline lesions characteristics-Evaluable subjects (N = 156)** | |
| **Primary infecting species (by Nested RT-PCR)** | |
| *L. (V.) braziliensis* | 106 |
| *L. (V.) guyanensis* | 7 |
| *L. (V.) lainsoni* | 4 |
| *L. (V.) peruviana* | 21 |
| Not identified | 15 |
| Negative for cutaneous leishmaniasis | 3 |
| **Body sites** | |
| Leg | 64 |
| Arm | 36 |
| Hand | 15 |
| Face | 15 |
| Back | 10 |
| Chest | 7 |
| Foot | 5 |
| Buttock | 2 |
| Head | 1 |
| Abdomen | 1 |
| **Estimated Days before sample collection that lesions were first noticed (days)** | |
| Mean (SD) | 35.7 (17.1) |
| Median | 33 |
| Range | 3–61 |

All lesions were ulcerated, none were manipulated, or had cellulitis.

The modified CL *Detect* Rapid Test sensitivity when compared with microscopy was 64.2% when the sample was collected using the dental broach but was higher at 83.3% when collected by lancet (Table 3). However, the false positive rate was higher with lancet specimens at 22.2% compared with 8.3% with dental broach specimens, making the specificity at 77.8% for scraping specimens and 91.7% for dental broach specimens (Table 3).

## Comparison of microscopy with kDNA-PCR

The modified *CL Detect Rapid Test* results versus the combined results of microscopy and the more sensitive kDNA-PCR method were examined. To compare the results, 2 x 2 tables were prepared for modified CL *Detect* Rapid Test positive and negative subjects by type of sample collection method (Tables 4 and 5). Of the 77 positive dental broach samples that were positive

**Table 2. Modified CL *Detect* Rapid Test vs Microscopy Test.**

| Modified *CL Detect* Rapid Test | Microscopy test | | | | | |
|---|---|---|---|---|---|---|
| | Dental broach specimens | | | Scraping by lancet specimens | | |
| | Positive | Negative | Total | Positive | Negative | Total |
| Positive | 77 | 3 | 80 | 100 | 8 | 108 |
| Negative | 43 | 33 | 76 | 20 | 28 | 48 |
| Total | 120 | 36 | 156 | 120 | 36 | 156 |

by microscopy, all were also positive by kDNA-PCR (Table 4). Seventy-six of the dental broach samples were negative, of which 43 were positive by microscopy and kDNA-PCR, and 73 were positive by kDNA-PCR. This means that 30 samples that were kDNA-PCR positive were microscopy negative (Table 4). The greater sensitivity when samples were collected by lancet scraping compared with the dental broach is supported by the kDNA-PCR data, where 108 samples were positive by modified CL *Detect* Rapid Test and kDNA-PCR, of which 8 of these samples were negative by microscopy (Table 5). These 8 samples were considered false positive when compared with microscopy but were confirmed positive by kDNA-PCR. The lower sensitivity of the microscopy method compared with kDNA-PCR is evident in 25 samples negative for leishmaniasis by both the microscopy method and the modified *CL Detect Rapid Test* with scraping-lancet that were positive by kDNA-PCR (S1 Table).

## Sensitivity and specificity Modified CL *Detect* vs microscopy by species

Of the two major species of leishmaniasis in Peru, *L. (V.) braziliensis* and *L. (V.) peruviana* were most frequently represented in the study population with 106 *L. (V.) braziliensis* samples and 21 *L. (V.) peruviana* samples (Table 4). Other species included *L. (V.) guyanensis* (7 samples), *L. (V.) lainsoni* (4 samples), and 15 kDNA-PCR positive samples that were not identifiable. The sensitivity and specificity of the modified *CL Detect Rapid Test* (microscopy as the standard comparator) by species for samples collected with the dental broach or by lancet scraping are shown in Table 4. As expected from the overall data, the sensitivity was higher for most of the species (with only 4 *L. (V.) lainsoni* samples they were the same) when samples were collected by lancet scraping compared with the dental broach. The sensitivity was 86.5% for *L. (V.) braziliensis*, 76.5% for *L. (V.) peruviana*, and 100% for *L. (V.) guyanensis* for lancet scraping (Table 4). Likewise, the specificity was lower (Table 4). Interestingly, the numbers of true positives for kDNA-PCR samples whose species was not identified by FRET-based Nested Real-Time PCR were 0 for dental broach samples and 1 for scraping samples.

**Table 3. Modified CL *Detect* Rapid Test Sensitivity and Specificity Analysis.**

| Modified CL Detect Rapid Test | N = 156 | |
|---|---|---|
| | Dental Broach Specimens | Scraping by Lancet Specimens |
| True Positives n (%) | 77 (49.4%) | 100 (64.1%) |
| False Negatives n (%) | 43 (27.6%) | 20 (12.8%) |
| True Negatives n (%) | 33 (21.2%) | 28 (17.9%) |
| False Positives n (%) | 3 (1.9%) | 8 (5.1%) |
| Sensitivity (95% CI) | 64.2% (54.9% - 72.7%) | 83.3% (75.4% - 89.5%) |
| Specificity (95% CI) | 91.7% (77.5% - 98.2%) | 77.8% (60.8% - 89.9%) |
| False positive rate (95% CI) | 8.3% (1.8% - 22.5%) | 22.2% (10.1% - 39.2%) |
| False negative rate (95% CI) | 35.8% (27.3% - 45.1%) | 16.7% (10.1% - 24.6%) |

**Table 4. Modified CL *Detect* Rapid Test Sensitivity and Specificity by Species of Samples Collected with Dental Broach–All Subjects.**

| Species | N | True Positive | False Negative | True Negative | False Positive | Sensitivity | Specificity |
|---|---|---|---|---|---|---|---|
| | | n (%) | n (%) | n (%) | n (%) | | |
| ***L. (V.) braziliensis*** | **106** | **61 (57.5)** | **28 (26.4)** | **14 (13.2)** | **3 (2.8)** | **68.5%** | **82.4%** |
| *L. (V.) peruviana* | 21 | 9 (42.9) | 8 (38.1) | 4 (19.0) | 0 (0.0) | 52.9% | 100.0% |
| *L. (V.) guyanensis* | 7 | 5 (71.4) | 2 (28.6) | 0 (0.0) | 0 (0.0) | 71.4% | NA[a] |
| *L. (V.) lainsoni* | 4 | 2 (50.0) | 2 (50.0) | 0 (0.0) | 0 (0.0) | 50.0% | NA |
| Not identified[b] | 15 | 0 (0.0) | 3 (20.0) | 12 (80.0) | 0 (0.0) | 100.0% | 100.0% |

[a] NA–cannot be calculated as denominator is zero.

[b] Sample was positive by kDNA-PCR but the species could not be identified.

### Analysis of false positives and false negatives

There were 8 false positives by the modified CL Detect compared with microscopy when samples were collected by lancet scraping; however, these were all positive by the more sensitive kDNA-PCR. In contrast, there were only 3 false positives when samples were collected with the dental broach. The overall false negative rate with scraping samples was 16.7% and with dental broach samples it was 35.8% (Tables 3 and 2, respectively). Although there were only 4 *L. (V.) lainsoni* positive samples, 2 were false negative by the CL *Detect* Rapid Test (Tables 4 and 5) by either collection method.

### Amastigote densities and modified CL Detect Rapid Test line intensities

As shown in Fig 1, the intensity of the line in the CL *Detect* Rapid Test should correlates with the density of *Leishmania* amastigotes in samples. Bubble charts showing the amastigote density versus the *CL Detect* Rapid Test line intensities for dental broach samples and lancet scraping samples are shown in Fig 2. There was a positive slope in both charts but linear regression $R^2$ was low for both samples, but higher for lancet scraping samples. Amastigote densities were also examined for *Leishmania* species. Mean and median densities were highest for *L. (V.) braziliensis* and *L. (V.) peruviana* (Table 6).

## Discussion

The CL *Detect* Rapid Test has 510(k) premarketing approval and was designed to detect the presence of *Leishmani*a. The basis for the approval was from studies conducted in Tunisia where *L. (L.) major* is prevalent and the United States (as a specificity control site). The results of these studies showed that the test sensitivity was 100% in the Old World (Tunisia study) and the specificity was 96% (US study). The overall false positive rate for the combined studies was

**Table 5. Modified *CL Detect* Rapid Test Sensitivity and Specificity by Species of Samples Collected by Lancet–All Subjects.**

| Species | N | True Positive n (%) | False Negative n (%) | True Negative n (%) | False Positive n (%) | Sensitivity | Specificity |
|---|---|---|---|---|---|---|---|
| *L. (V.)braziliensis* | 106 | 77 (72.6) | 12 (11.3) | 11 (10.4) | 6 (5.7) | 86.5% | 64.7% |
| *L. (V.)peruviana* | 21 | 13 (61.9) | 4 (19.0) | 3 (14.3) | 1 (4.8) | 76.5% | 75.0% |
| *L. (V.)guyanensis* | 7 | 7 (100.0) | 0 (0.0) | 0 (0.0) | 0 (0.0) | 100% | NAa) |
| *L. (V.) lainsoni* | 4 | 2 (50.0) | 2 (50.0) | 0 (0.0) | 0 (0.0) | 50.0% | NA |
| Not identifiedb) | 15 | 1 (6.7) | 2 (13.3) | 11 (73.3) | 1 (6.7) | 33.3% | 91.7% |

a) NA–cannot be calculated as denominator is zero.

b) Sample was positive by PCR, but the species could not be identified.

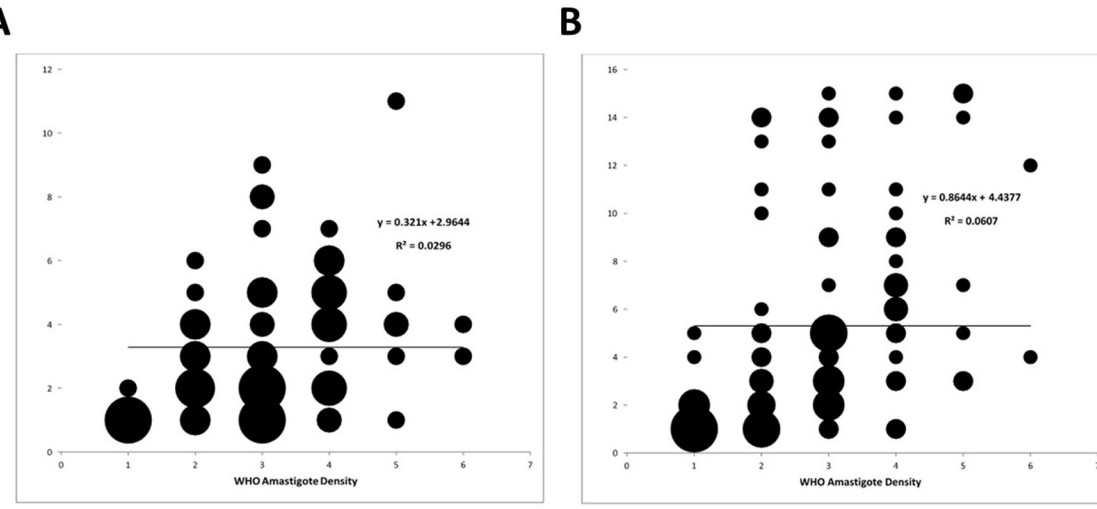

**Fig 2. Modified CL *Detect* Rapid Test Intensity versus WHO Amastigote Density of Samples Collected by Scraping.** The figure shows bubble charts for dental broach (A) and lancet scrapping (B). The Y-axis shows the CL-detect signal intensity (scale 0 to 15) whereas the X-axis shows the WHO amastigote density (scale 1–6).

5.3% [3]. However, post approval studies have shown lower sensitivities, especially in the New World [4,12] and in Ethiopian patients suspected of CL caused by *Leishmania aethiopica*.

The FDA approved kit uses specimens collected with a dental broach. This procedure is painful, invasive and demanding when compared to the more common practice to collect samples by scraping, the standard sample collecting method for CL diagnosis worldwide. Use of a dental broach for sample collection usually requires an injection of a local anesthetic for numbing during the sampling procedure. Local anesthesia not only takes time, but entails all the occasionally side effects of an injectable anesthetic especially in children that cannot easily tolerate the pain. The improvement of the sensitivity of the test by the scraping method compared to the more invasive and complicated use of the dental broach is important and can also be explained by the amount of CL lesion tissue sample collected. An increase in the amount of tissue results in a higher number of amastigotes and the amount target antigen present increasing the test sensitivity of the test. This study confirmed that by changing the assay conditions that the overall sensitivity was improved to 64.2% from 30% (12) found under the label assay conditions in the New World.

A further improvement was observed when the samples were collected by lancet scraping at a sensitivity of 83.3% with the sensitivity for *L. (V.) braziliensis* at 86.5% (S2 Table). However, there was a loss in specificity at 77.8% overall and at 64.7% for *L. (V.) braziliensis* with a false positive rate at 22.2% overall and at 35.3% *for L. (V.) braziliensis*. However, all 8 false negative lancet scraping samples were positive by kDNA-PCR. Thus, samples positive by the modified

**Table 6. WHO Amastigote Density Scale in Microscopy Positive Specimens.**

| Species | N | Mean | SD | Median | Min | Max |
|---|---|---|---|---|---|---|
| *L. (V.) braziliensis* | 89 | 2.7 | 1.3 | 3 | 1 | 6 |
| *L. (V.) peruviana* | 17 | 2.6 | 0.8 | 3 | 1 | 4 |
| *L. (V.) guyanensis* | 7 | 1.9 | 0.9 | 2 | 1 | 3 |
| *L. (V.) lainsoni* | 4 | 2.5 | 2.4 | 1.5 | 1 | 6 |
| Not identified[a] | 3 | 1 | 0 | 1 | 1 | 1 |

[a] Sample was positive by kDNA-PCR but the species could not be identified

CL *Detect* Rapid Test but negative by microscopy were in fact from subjects infected with *Leishmania*. Therefore, the decision to start treatment of a patient based on a positive result from the modified CL *Detect* Rapid Test alone is reasonable until such time as further diagnostic tests can be performed.

Results from the modified CL *Detect* are available within 30 minutes, a significant improvement over the current process generally used in endemic countries. Diagnosing CL often requires samples to be shipped to and tested at reference central laboratories far from where the patients reside and taking days if not weeks to receive test results. This is important because in many endemic countries like Peru, CL treatment is free but is not provided by the Ministry of Health without a positive diagnostic test. Rapid diagnosis means treatment can begin immediately, people do not have to leave their workplace multiple times, and travel, sometimes times for days, to visit the treatment center.

For the Department of Defense of the United States, the modified CL *Detect* will serve as an important asset in the general diagnosis of skin lesions, distinguishing cutaneous leishmaniasis from other important infectious diseases with a similar presentation. It will allow implementation of appropriate treatments, which can reduce the severity of scarring, reduce lost duty time and improve healthcare and morale for U.S. military personnel in numerous areas of operations. Deployed medics will be able to diagnose CL in 98 countries where CL is endemic, including New World species.

A limitation of this study has been the inability of comparing modified vs non-modified CL Detect, although the study design allows determining the sensitivity and specificity of the modified CL Detect with microscopy and PCR as references the objective if this study and what we wanted to evaluate. Another limitation of this study is that PERU-1 and PERU-2 use different patients, therefore direct comparison is not possible. In accordance with the InBios insert only ulcerative lesions tested were less than four months old, Mean (SD) 35.7 (17.1) days old, these are the type of lesions that have higher parasite load, then perhaps the sensitivity is lower in lesions older than two months. Notwithstanding, the advantage of the CL Detect is that is really a point-of-care rapid test and can be used where the patients are in most cases in the field, allowing early detection of cases.

## Conclusion

Our study shows two modifications to the CL Detect package insert, increasing the extraction time of the tissue sample and increasing the volume of the sample added to the test strip increased the diagnostic sensitivity of the test to 83.3% comparable to microscopy for *Leishmania* species found in Peru. This point-of-care test may enable earlier anti-leishmanial drug treatment decisions based on a positive result from the CL Detect Rapid Test alone until further diagnostic tests like microscopy and PCR can be performed. When available the high specificity of direct microscopy and the easy-to-use CL Detect can be performed from the same commonly used scraping sample to lessen the number of diagnostic cases where amastigotes are not found, and number of tissue samples needed. Nonetheless, in our opinion based on published results [7,8,9,12] and our own data there is still a need for improvement to increase the sensitivity and specificity. The need for a point-of-care rapid test that could be used in all type of lesions (ulcerated and non-ulcerated), lesion of all durations (ages) to supply diagnostic capabilities to the most vulnerable populations in Peru and most endemic countries remains high.

## Supporting information

**S1 File. Supplemental Methods.**
(DOCX)

**S2 File. ISSM_STARD_Checklist.**
(PDF)

**S3 File. STARD diagram.**
(DOCX)

**S1 Table. Microscopy by PCR for CL *Detect* Rapid Test (Collected by Dental Broach and Collected by Scraping) Positive and negative Subjects.**
(DOCX)

**S2 Table. Comparison of Results of All Studies by Different Assay Conditions.**
(DOCX)

## Acknowledgments

We are indebted to the patients and Ministry of Health of the Madre de Dios region of Peru, the nurses, Silvia Revilla and Gloria Fuentes, and scientists, Dr. Danette Bishop and Dr. Edward Smith who contributed to the study as well as to Dr. Joan L. Cmarik, Science Advisor/ DHP 6.7 Manager, Office of the Principal Assistant for Acquisition US Army Medical Center and Development Command, for the excellent management of D6.7_16_C2_I_16_J9_1527F and support to the project and NAMRU-6. We are also grateful to InBios International Inc., USA for the donation of the CL Detect Rapid Test. The company had no role in study design, data collection and analysis, decision to publish, or preparation of the manuscript.

## Copyright statement

Some authors of this manuscript are military service members and employees of the U.S. Government. This work was prepared as part of their official duties. Title 17 U.S.C. §105 provides that "Copyright protection under this Title is not available for any work of the United States Government". Title 17 U.S.C. §101 defines a U.S. Government work as a work prepared by a military service member or employee of the U.S. Government as part of that person's official duties.

## Disclaimer

The views expressed in this manuscript reflect the results of research conducted by the author and do not necessarily reflect the official policy or position of the Department of the Navy, Department of Defense, nor the United States Government

## Author Contributions

**Conceptualization:** Max Grogl, Ngami Donovan, Janet H. Ransom, Ana Ramos, Elmer Llanos Cuentas.

**Data curation:** Max Grogl, Christie A. Joya, Maria Saenz, Rocio del Pilar Santos, Maxy B. De los Santos, Ngami Donovan, Janet H. Ransom, Ana Ramos, Elmer Llanos Cuentas.

**Formal analysis:** Max Grogl, Christie A. Joya, Rocio del Pilar Santos, Maxy B. De los Santos, Ngami Donovan, Janet H. Ransom, Ana Ramos, Elmer Llanos Cuentas.

**Investigation:** Max Grogl, Christie A. Joya, Ana Quispe, Luis Angel Rosales, Rocio del Pilar Santos, Maxy B. De los Santos, Ngami Donovan, Janet H. Ransom, Ana Ramos, Elmer Llanos Cuentas.

**Methodology:** Max Grogl, Maria Saenz, Rocio del Pilar Santos, Maxy B. De los Santos, Ngami Donovan, Janet H. Ransom, Ana Ramos, Elmer Llanos Cuentas.

**Project administration:** Max Grogl, Christie A. Joya, Maria Saenz.

**Supervision:** Max Grogl, Christie A. Joya.

**Validation:** Max Grogl, Rocio del Pilar Santos, Maxy B. De los Santos, Ngami Donovan, Janet H. Ransom, Ana Ramos, Elmer Llanos Cuentas.

**Visualization:** Max Grogl, Christie A. Joya, Maria Saenz, Maxy B. De los Santos.

**Writing – original draft:** Max Grogl, Christie A. Joya, Ngami Donovan, Janet H. Ransom, Ana Ramos, Elmer Llanos Cuentas.

**Writing – review & editing:** Max Grogl, Christie A. Joya, Ana Quispe, Luis Angel Rosales, Rocio del Pilar Santos, Maxy B. De los Santos, Ngami Donovan, Janet H. Ransom, Ana Ramos, Elmer Llanos Cuentas.

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
