## [Decision Letter · Decision Letter 0]

12 Jul 2022

Dear Dr. Joya,

Thank you very much for submitting your manuscript "Evaluation of a diagnostic device, CL Detect™ Rapid Test for the diagnosis of new world cutaneous leishmaniasis" for consideration at PLOS Neglected Tropical Diseases. As with all papers reviewed by the journal, your manuscript was reviewed by members of the editorial board and by several independent reviewers. In light of the reviews (below this email), we would like to invite the resubmission of a significantly-revised version that takes into account the reviewers' comments. 

In particular, a major concern raised by the reviewers is that the validity of the assay in terms of sensitivity and specificity is not high enough to be used as a routine rapid diagnostic test for the species of the New World. The limitation of this method should be discussed in the manuscript. 

We cannot make any decision about publication until we have seen the revised manuscript and your response to the reviewers' comments. Your revised manuscript is also likely to be sent to reviewers for further evaluation.

Sincerely,

Alberto Novaes Ramos Jr

Academic Editor

Laura-Isobel McCall

Section Editor

Relevant additional considerations by the reviewers:

The validity of the assays in terms of sensitivity and specificity is not well enough to be used as a routine rapid diagnostic test for the species of the New World. The original test was developed for Leishmania major with superior validity values. The limitation of this approach must necessarily be considered by the authors for the paper to have consistency - the findings presented indicate that it is not going to be a valid test for the New World's species.

Reviewer's Responses to Questions

**Key Review Criteria Required for Acceptance?**

**Methods**

-Are the objectives of the study clearly articulated with a clear testable hypothesis stated?

-Is the study design appropriate to address the stated objectives?

-Is the population clearly described and appropriate for the hypothesis being tested?

-Is the sample size sufficient to ensure adequate power to address the hypothesis being tested?

-Were correct statistical analysis used to support conclusions?

-Are there concerns about ethical or regulatory requirements being met?

Reviewer #1: The methods used in this research are acceptable in terms of feasibility, sample size, statistical analysis, etc.

Reviewer #2: This is a clear and relevant study for researchers and practioners working with cutaneous leishmaniasis in the New World. The methodology is clearly described and appropriate and sample size adequate for the methodology tested.

Reviewer #3: Although statistical methods were mentioned no statistical analysis between different approaches was not performed. The authors simply stated that the sensitivity was higher in specimens obtained (83.3%) than those from dental broach (64.2%) and did not mention the significance of the statistical difference.

Reviewer #4: Specific Comments : 

>>> Please check point by point (as possible) the comments included in the Word file 

Thank you

Reviewer #5: (No Response)

**Results**

-Does the analysis presented match the analysis plan?

-Are the results clearly and completely presented?

-Are the figures (Tables, Images) of sufficient quality for clarity?

Reviewer #1: The analysis presented match the analysis plan and results, images and graphs are clearly presented.

Reviewer #2: The results a clearly presented.

Reviewer #3: The significance level of different methods should be performed and the p-values should be presented.

Reviewer #4: Specific Comments : 

>>> Please check point by point (as possible) the comments included in the Word file 

Thank you

Reviewer #5: (No Response)

**Conclusions**

-Are the conclusions supported by the data presented?

-Are the limitations of analysis clearly described?

-Do the authors discuss how these data can be helpful to advance our understanding of the topic under study?

-Is public health relevance addressed?

Reviewer #1: The discussion section needs to be completed. Other similar studies can also be added in this section.

Reviewer #2: Conclusions are supported by presented data, discussion and relevance of the methodology is well addressed.

Reviewer #3: The conclusions are supported by the data but the limitation of the analysis and public health relevance has not been addressed

The discussion section is very weak. In this section, the sensitivity of the conventional methods such as direct examination and culture media should be reviewed. The weakness of the test including the detection of the organism to the genus level should be described. Also, some critical factors which play some roles in influencing the sensitivity of the assay should be mentioned.

Reviewer #4: The conclusions are supported by the presented data 

There is a need of introducing a limitation section 

Yes 

Yes

Reviewer #5: (No Response)

**Editorial and Data Presentation Modifications?**

Reviewer #1: (No Response)

Reviewer #2: I suggest a correction on page 9 line 181 related to Leishmania strains: L.(V) amazonensis is written as V. (Viannia) , should be (L.) of Leishmania and also for L (L.) mexicana.

Reviewer #3: --

Reviewer #4: Specific Comments : 

>>> Please check point by point (as possible) the comments included in the Word file 

Thank you

Reviewer #5: (No Response)

**Summary and General Comments**

Reviewer #1: In my opinion, considering the changes that have been made by the researchers to perform this quick test and have achieved significant results, it is a positive point that can be useful.

Reviewer #2: This is an interesting study and with the modification suggested by the authors this could be a new kit for diagnosis of ACL.

Reviewer #3: The authors used a modification of the cutaneous leishmaniasis Detection Rapid Test for the diagnosis of New World CL species. This device has already been tested for Leishmania major, the causative agent of the Old Word zoonotic leishmaniasis. However, the sensitivity here was significantly lower by scraping and also dental broach. This method is qualitative, in vitro immunochromatographic assay for the rapid detection of Leishmania spp. antigen in ulcerative skin lesions. However major revisions are needed before being considered for publication.

Reviewer #4: Following the PLOSNTD reviewers standards questions, I start with a general appreciation before going to some specific comments in a Word attached file

General Comments : 

What are the main claims of the paper and how significant are they for the discipline?

>>> This is a very interesting study attending to complete the gap of existing evidence about the CL Detect rapid test in the context of New World endemic CL areas. 

Are these claims novel? If not, which published articles weaken the claims of originality of this one?

>>> The originality is linked to the study done in the specific context of PERU where the study was done and where many CL species from the new world could be targeted 

Are the claims properly placed in the context of the previous literature? Have the authors treated the literature fairly?

>>> Some additional literature are suggested to be added to complete the view 

Do the data and analyses fully support the claims? If not, what other evidence is required?

>>> The method explained and followed is associated to good presentation of data and enough analyses to give authors research questions answers 

Would additional work improve the paper? How much better would the paper be if this work were performed and how difficult would it be to do this work?

>>> Some comments in the word file could be taken by the authors to enhance the next version of this manuscript. The final decision as you know depend on the Editor. 

PLOS Neglected Tropical Diseases encourages authors to publish detailed protocols and algorithms as supporting information online. Do any particular methods used in the manuscript warrant such treatment? 

>>> The authors noticed the previous authorisations and the protocol is part of the formal administrative and governmental authorisation given in 2018. 

>>> The ethical consideration are all well presented and followed

If a protocol is already provided, for example for a randomized controlled trial, are there any important deviations from it? If so, have the authors explained adequately why the deviations occurred?

>>> The only suggestion is to use the STARD checklist than the CONSORT checklist because the study design is not a clinical trial one

Is this paper outstanding in its discipline? If yes, what makes it outstanding? If not, why not?

>>> For my side, this manuscript is very interesting and will contribute to many other opportunities for further research based on very robust research questions on the field. The potential to be cited many times in the near future is obvious. 

If the paper is considered unsuitable for publication in its present form, does the study itself show sufficient potential that the authors should be encouraged to resubmit a revised version?

>>> Yes, the revised version would be very interesting 

Are original data deposited in appropriate repositories and accession/version numbers provided for genes, proteins, mutants, diseases, etc.?

>>> N/A

Are details of the methodology sufficient to allow the experiments to be reproduced?

>>> Yes 

Is the manuscript well organized and written clearly enough to be accessible to non-specialists?

>>> Yes

Reviewer #5: (No Response)

PLOS authors have the option to publish the peer review history of their article (what does this mean?). If published, this will include your full peer review and any attached files.

Reviewer #1: Yes: Dr. Pooya Ghasemi Nejad Almani

Reviewer #2: No

Reviewer #3: No

Reviewer #4: No

Reviewer #5: No
---

## [Decision Letter · Decision Letter 1]

6 Oct 2022

Dear Dr. Joya,

Thank you very much for submitting your manuscript "Evaluation of a diagnostic device, CL Detect Rapid Test for the diagnosis of new world cutaneous leishmaniasis in Peru" for consideration at PLOS Neglected Tropical Diseases. As with all papers reviewed by the journal, your manuscript was reviewed by members of the editorial board and by several independent reviewers. The reviewers appreciated the attention to an important topic. Based on the reviews, we are likely to accept this manuscript for publication, providing that you modify the manuscript according to the review recommendations. 

Sincerely,

Alberto Novaes Ramos Jr

Academic Editor

Laura-Isobel McCall

Section Editor

Reviewer's Responses to Questions

**Key Review Criteria Required for Acceptance?**

**Methods**

-Are the objectives of the study clearly articulated with a clear testable hypothesis stated?

-Is the study design appropriate to address the stated objectives?

-Is the population clearly described and appropriate for the hypothesis being tested?

-Is the sample size sufficient to ensure adequate power to address the hypothesis being tested?

-Were correct statistical analysis used to support conclusions?

-Are there concerns about ethical or regulatory requirements being met?

Reviewer #1: The materials and methods have the necessary standards.

Reviewer #3: - The authors should define the Leishmania species that have initially worked with when stating the objective.

- The blade used to collect samples by the lancet should be specified.

- Describe the exact procedure for taking the tissue scraping.

Reviewer #4: (No Response)

**Results**

-Does the analysis presented match the analysis plan?

-Are the results clearly and completely presented?

-Are the figures (Tables, Images) of sufficient quality for clarity?

Reviewer #1: Results and figures are clearly presented.

Reviewer #3: Yes

Reviewer #4: (No Response)

**Conclusions**

-Are the conclusions supported by the data presented?

-Are the limitations of analysis clearly described?

-Do the authors discuss how these data can be helpful to advance our understanding of the topic under study?

-Is public health relevance addressed?

Reviewer #1: the conclusions supported are by the data presented

Reviewer #3: Yes

Reviewer #4: (No Response)

**Editorial and Data Presentation Modifications?**

Reviewer #1: In my opinion, the article is acceptable.

Reviewer #3: (No Response)

Reviewer #4: (No Response)

**Summary and General Comments**

Reviewer #1: In this work, the authors have modified a method that increases the ability to detect leishmaniasis in endemic areas with indigenous species, so it is significant.

Reviewer #3: - Justification for performing such time-consuming work should be clarified. This technique is not a rapid and easy approach to the selection of proper and timely therapy. Conversely, it is a tedious, labor-intensive, cumbersome, and costly procedure . In reality, any conventional diagnostic method should be performed quickly with acceptable sensitivity. We should not neglect simple conservative methods which are achieved through multiple global efforts to be replaced by frustrating, complicated, over-priced, and non-applicable approaches. These techniques are not pertinent and recommended in remote areas of tropical regions with low-income countries where this complex disease is present.

- The authors should present the background information about the Leishmania spp. that causes CL in the area. As the authors well pointed out, more than 20 Leishmania species are causative. Most of them cause CL.

- - L 67, 350 million people at risk should be over1 billion…(WHO-Leishmaniasis home).

- L64, for cutaneous leishmaniasis, presents the abbreviation form (CL) and excludes the CL abbreviation in L72.

- In vitro should be italicized (L69 and other places).

- L94 and many places Leishmania species should be italicized.

- In the paragraph L16-121, there are several techniques used to detect the Leishmania organism, but depending on the species, the lesion sampling stage, personnel, and supplies the sensitivity of these tests are diverse.

Reviewer #4: (No Response)

PLOS authors have the option to publish the peer review history of their article (what does this mean?). If published, this will include your full peer review and any attached files.

Reviewer #1: Yes: Pooya Ghasemi Nejad Almani

Reviewer #3: No

Reviewer #4: No

Figure Files:

Data Requirements:

Reproducibility:

References

---

## [Decision Letter · Decision Letter 2]

28 Nov 2022

Dear Dr. Joya,

Thank you very much for submitting your manuscript "Evaluation of a diagnostic device, CL Detect Rapid Test for the diagnosis of new world cutaneous leishmaniasis in Peru" for consideration at PLOS Neglected Tropical Diseases. As with all papers reviewed by the journal, your manuscript was reviewed by members of the editorial board and by several independent reviewers. The reviewers appreciated the attention to an important topic. Based on the reviews, we are likely to accept this manuscript for publication, providing that you modify the manuscript according to the review recommendations. 

Specifically, please make sure to provide the STARD checklist and flow diagram. updated for this final version of the manuscript.

Sincerely,

Alberto Novaes Ramos Jr

Academic Editor

Laura-Isobel McCall

Section Editor

Reviewer's Responses to Questions

**Key Review Criteria Required for Acceptance?**

**Methods**

-Are the objectives of the study clearly articulated with a clear testable hypothesis stated?

-Is the study design appropriate to address the stated objectives?

-Is the population clearly described and appropriate for the hypothesis being tested?

-Is the sample size sufficient to ensure adequate power to address the hypothesis being tested?

-Were correct statistical analysis used to support conclusions?

-Are there concerns about ethical or regulatory requirements being met?

Reviewer #1: The materials and methods section clearly explains the research process.

Reviewer #3: Different surgical blades could be used to obtain the sample. The authors should specify the number of blades used for this purpose (L148).

Reviewer #4: The authors made consistent efforts to enhance their manuscript. 

Then, the appraisal of all manuscript quality it could be completed by adding the STARD checklist. Because, in the newest revision form the authors discarded another version of that checklist that was not too suitable to assess all the components. My last suggestion is to use the original 2015 STARD checklist that is available from

https://www.equator-network.org/reporting-guidelines/stard/

Also, one statement of using this checklist should be included in the method section,

**Results**

-Does the analysis presented match the analysis plan?

-Are the results clearly and completely presented?

-Are the figures (Tables, Images) of sufficient quality for clarity?

Reviewer #1: The analysis of the results is done correctly and the clear images show the results.

Reviewer #3: OK

Reviewer #4: To summarize the results, the STARD Flow diagram at the beginning of the results section is also needed (available from the same link : https://www.equator-network.org/reporting-guidelines/stard/)

**Conclusions**

-Are the conclusions supported by the data presented?

-Are the limitations of analysis clearly described?

-Do the authors discuss how these data can be helpful to advance our understanding of the topic under study?

-Is public health relevance addressed?

Reviewer #1: The conclusion part also seems complete

Reviewer #3: OK

Reviewer #4: (No Response)

**Editorial and Data Presentation Modifications?**

Reviewer #1: (No Response)

Reviewer #3: The scientific name of the organisms should be written in italic font (L94 & 408) 

- The author should use the abbreviation for the commonly used terms ( cutaneous leishmaniasis L51, 57, 61, 72, 80, 387).

Reviewer #4: (No Response)

**Summary and General Comments**

Reviewer #1: (No Response)

Reviewer #3: OK

Reviewer #4: (No Response)

PLOS authors have the option to publish the peer review history of their article (what does this mean?). If published, this will include your full peer review and any attached files.

Reviewer #1: Yes: Dr. Pooya Ghasemi Nejad Almani

Reviewer #3: No

Reviewer #4: No

Figure Files:

Data Requirements:

Reproducibility:

References

---

## [Decision Letter · Decision Letter 3]

21 Dec 2022

Dear Dr. Joya,

We are pleased to inform you that your manuscript 'Evaluation of a diagnostic device, CL Detect Rapid Test for the diagnosis of new world cutaneous leishmaniasis in Peru' has been provisionally accepted for publication in PLOS Neglected Tropical Diseases.

Best regards,

Alberto Novaes Ramos Jr

Academic Editor

Laura-Isobel McCall

Section Editor

All minor issues brought up by reviewer 3 should be considered by the authors in the final version of the manuscript.

1- In stating the objective one should describe the Leishmania species which is used to evaluate the diagnostic device, CL Detect Rapid Test for the diagnosis of new world cutaneous leishmaniasis in Peru, once in the abstract (L25-31) and another in the introduction (L125-132).

2- Leishmania species should be written in italic font in L94 & 411.

3- The abbreviation form of cutaneous leishmaniasis should be used throughout the manuscript (L72, 80, 390). The authors have previously defined the abbreviated form but it is not employed properly.

Reviewer's Responses to Questions

**Key Review Criteria Required for Acceptance?**

**Methods**

-Are the objectives of the study clearly articulated with a clear testable hypothesis stated?

-Is the study design appropriate to address the stated objectives?

-Is the population clearly described and appropriate for the hypothesis being tested?

-Is the sample size sufficient to ensure adequate power to address the hypothesis being tested?

-Were correct statistical analysis used to support conclusions?

-Are there concerns about ethical or regulatory requirements being met?

Reviewer #3: Thanks for the corrections made. I realize it is frustrating, but have to resolve the problem consistent with the journal’s style. The following previous revisions should be carefully corrected to prevent elongation of the decision:

1- In stating the objective one should describe the Leishmania species which is used to evaluate the diagnostic device, CL Detect Rapid Test for the diagnosis of new world cutaneous leishmaniasis in Peru, once in the abstract (L25-31) and another in the introduction (L125-132).

2- Leishmania species should be written in italic font in L94 & 411.

3- The abbreviation form of cutaneous leishmaniasis should be used throughout the manuscript (L72, 80, 390). The authors have previously defined the abbreviated form but it is not employed properly.

Reviewer #4: (No Response)

**Results**

-Does the analysis presented match the analysis plan?

-Are the results clearly and completely presented?

-Are the figures (Tables, Images) of sufficient quality for clarity?

Reviewer #3: See above.

Reviewer #4: (No Response)

**Conclusions**

-Are the conclusions supported by the data presented?

-Are the limitations of analysis clearly described?

-Do the authors discuss how these data can be helpful to advance our understanding of the topic under study?

-Is public health relevance addressed?

Reviewer #3: See above.

Reviewer #4: (No Response)

**Editorial and Data Presentation Modifications?**

Reviewer #3: See above.

Reviewer #4: (No Response)

**Summary and General Comments**

Reviewer #3: See above.

Reviewer #4: (No Response)

PLOS authors have the option to publish the peer review history of their article (what does this mean?). If published, this will include your full peer review and any attached files.

Reviewer #3: **Yes: **Iraj Sharifi

Reviewer #4: No

---

## [Editor Report · Acceptance letter]

24 Feb 2023

Dear Dr. Joya,

We are delighted to inform you that your manuscript, "Evaluation of a diagnostic device, CL Detect Rapid Test for the diagnosis of new world cutaneous leishmaniasis in Peru," has been formally accepted for publication in PLOS Neglected Tropical Diseases.

Best regards,

Shaden Kamhawi

co-Editor-in-Chief

Paul Brindley

co-Editor-in-Chief
